# Models for the Evaluation of Productivity and Costs of Mechanized Felling on Poplar Short Rotation Coppice in Italy

**Giulio Sperandio \*** , **Andrea Acampora** , **Angelo Del Giudice and Vincenzo Civitarese**

Consiglio per la Ricerca in Agricoltura e L'analisi Dell'economia Agraria (CREA)—Centro di Ricerca Ingegneria e Trasformazioni Agroalimentari, via della Pascolare 16, 00015 Monterotondo, Italy; andrea.acampora@crea.gov.it (A.A.); angelo.delgiudice@crea.gov.it (A.D.G.); vincenzo.civitarese@crea.gov.it (V.C.)
\* Correspondence: giulio.sperandio@crea.gov.it

**Abstract:** The forest biomass, as a renewable energy source, can significantly contribute to the progressive replacement of fossil fuels in energy production, with a positive final balance in terms of greenhouse gas emissions. One of the different sources of woody biomass supply is represented by short rotation coppices (SRC) plantations, currently present in various European countries for a total of about fifty thousand hectares. In Italy, part of the SRC surface has been converted into other more profitable crops, both the low levels reached by the woodchips market price and the scarce availability of specific public incentives. In this study, the authors expose the results of the models for evaluating work time, productivity, and costs of the felling operation on SRC poplar plantations with 8- and 11-year-old trees. The aim is to evaluate the economic sustainability in the use of advanced mechanization on these plantations. The machine was a crawler excavator equipped with a shear head. In the 11-year-old plantation, the productivity estimation model returned a range of 1.09–18.93 Mg h$^{-1}$ (average 5.56 $\pm$ 3.88 SD) when the weight variation of the trees was 20–491 kgw (average 100.41 $\pm$ 87.48 SD). In the 8-year-old poplar, the range was 1.02–11.60 Mg h$^{-1}$ (average 3.80 $\pm$ 1.71 SD), for weight variation of 17–137 kgw (average 50.57 $\pm$ 18.82 SD). The consequent variation in unit cost was EUR 2.82–51.63 Mg$^{-1}$ and EUR 4.05–49.65 Mg$^{-1}$, corresponding to EUR 1252.17–3463.78 ha$^{-1}$ and EUR 922.49–2545.11 ha$^{-1}$ for 11- and 8-year-old trees, respectively.

**Keywords:** felling; shear head; poplar; SRC; time study; productivity; costs

## 1. Introduction

In the debate on the production of energy from renewable sources, woody biomass plays an important role, representing almost 70% of the total European renewable energies [1]. In the future, in accordance with the energy policy objectives at the European level, the demand for biomass should increase to meet the needs of the energy production sector. This growing demand involves fuelwood and higher-quality energy products [2,3], as well as wood building material or other biomaterials for different uses [4]. Greater use of biomass can concretely contribute to the progressive replacement of fossil fuels in energy production, with a positive final balance in terms of greenhouse gas emissions [5,6]. Among the various sources of woody biomass supply, the fast-growing short rotation woody crops (Short Rotation Coppices—SRC) could contribute significantly to the achievement of the 2020 European targets [7].

Currently, at the European level, about 50,000 ha of SRC are estimated, which represents a limited area considering the European and national incentives and political support measures in recent years in favor of these crops [8]. The European countries that pay greater attention to these plantations and have a greater surface invested in SRC are mainly Sweden, Poland, United Kingdom, Germany, and Italy [7]. In other European countries, these plantations are rather limited but there are plans and political wills to favor their development in the near future [1].



In Italy, in the last decade, there has also been a substantial downsizing of the SRC surface due to the strong critical position taken by public opinion regarding the atmospheric emissions produced by all biomass combustion processes [9]. However, the national production of thermal energy represents a relevant sector, and renewable sources can concretely contribute to the achievement of national energy and environmental goals in the medium–long term (2030–2050). This has led to a significant change in the technological development of the biomass plants with the adoption of technical solutions and energy processes, also developed on a small scale [10], that is more respectful of the environmental aspects [11]. The current biomass energy plants, in fact, are always equipped with efficient and technological filtering systems to effectively reduce pollution by fine dust [12].

SRC plantations have been an object of intense experimentation and cultivation over the past twenty years [13–15]. However, if we consider the current area covered by these crops, we can say that what is foreseen for these plantations has not been achieved. The maximum SRC area was reached a few years ago, when an overall national area of about 7000 ha was estimated, made using mainly poplar clones, followed by eucalyptus, willow, and black locust, most of which are located in Northern Italy [16,17]. This area has progressively decreased due to both the reduction in economic margins due to the low levels reached by the market price of the woodchips, and the decreased availability of specific public incentives. For these reasons, part of the existing area that has invested in SRC plantations has been converted to other more profitable types of crops [18].

Many of the remaining plantations, when not converted, have been partly abandoned or not managed according to the usual planned cuts, perhaps waiting for more favorable market conditions. The lengthening of the cutting shift influences the working system with which the plantation is harvested, determining the level and type of mechanization to be used [19]. Under normal plantation growth conditions, if the cutting is not done every two or three years, it will no longer be possible to use the modified forage harvester normally used on these crops [20,21]. In this case, when the diameter at the base of the tree exceeds 15–20 cm, it is necessary to resort to forest mechanization [22].

The introduction of forest mechanization techniques, such as feller-buncher heads, specialized tractors for logging, and processors for mechanized delimbing, began in the early 1980s in conifer plantations created for industrial purposes [23].

Currently, the use of advanced mechanization, such as harvesters and forwarders, is constantly increasing [24,25], especially on poplar [26] and eucalyptus [27], while it is not widespread in mountain forests compared to other European countries [28,29]. This type of mechanization could also be used on SRC, when the size of the tree makes the use of dedicated machines impossible. The harvesting method adopted in these cases is the whole tree system (WTS). This system involves cutting the whole tree or dividing it into two parts, depending on the machine, to be used for subsequent extraction up to the landing area, where the trees are subsequently chipped. WTS is generally applied in forestry yards using different levels of mechanization on harvesting operation, substantially represented by: (a) motor-manual harvesting, using a chainsaw for felling and tractor with grapple or winch for wood extraction; (b) partially mechanized harvesting, using a chainsaw for felling and skidding with a forwarder or skidder; or (c) fully mechanized harvesting, using a felling machine (shear, disc saw or chainsaw head) and forwarder or skidder with grapple for tree extraction [30].

In this study, the authors present the results of a research activity developed within the national AGROENER project. The purpose of this study was to evaluate the technical and economic sustainability in the use of advanced mechanization for felling of two SRC poplar plantations: the first with 8-year-old trees and the second with 11-year-old trees. Sustainability was assessed by comparison models of the work time, productivity, and costs between the two crop cycles.

## 2. Materials and Methods

### 2.1. SRC Poplar Plantations

The SRC poplar plantation is located inside the farm of CREA, Research Centre for Engineering and Agro-food Processing, in Monterotondo (Rome), Central Italy (42°06′4.40″ N, 12°37′31.41″ E). The plantation was planted in 2005 on a flat surface of 4.5 ha on a clayey-loamy soil with a low content of organic matter, nitrogen, and phosphorus [17,31]. The climatic conditions are typical of the Mediterranean area with cold winters and dry summers. The average temperature in the last ten years has been 16.2 °C, with an average annual rainfall of 821 mm in about 81 rainy days per year.

The plantation considered in the study consisted of clone AF2 (*Populus × canadensis* Moenech), with an initial density of 7140 plants ha$^{-1}$ and with a distance between the rows of 2.80 m and between the plants along the row of 0.50 m. The plantation was harvested in previous years with different harvesting methods. The previous harvesting operations were carried out in spring 2008 in one area and in 2011 in another area. The last cut was made in 2019 with the use of advanced mechanization consisting of a crawler excavator equipped with a shear head. The cutting area of the AF2 clone had a total area of 0.49 ha and was divided into two sections of seven rows each. In the first one, the trees were 8 years old (T8), while in the second one, they were 11 (T11). The roots, in both cases, were 14 years old. The T8 section was approximately 0.27 ha, with an average row length of 138 m, compared to 0.22 ha for the T11 section, with an average row length of 112 m. At the time of cutting, there was only one dominant tree for each live stump, also due to the old age of the trees. To highlight the differences in the operating performance of the machine and, consequently, in the costs of the operation, the felling work was observed separately for the two sections.

Before cutting, the diameters at breast height (DBH) and heights on each row were measured to determine the dendrometric characteristics of all the standing trees. A Mantax 500 dendrometric caliper (accuracy 1 mm) and a Trupulse R360 laser rangefinder (accuracy 0.10 m) were used, respectively. To determine the relationship between tree weight and DBH, a sample of 30 representative trees was taken for each section, and each tree was weighed with a Kern digital dynamometer (accuracy 100 g).

Equation (1) establishes the best relationship between weight and tree DBH:

$$W = a \times D^2 + b \times D \tag{1}$$

where $W$ represents the tree weight (kgw); $a$ and $b$ the equation coefficients and $D$ the DBH (cm).

The two equations obtained from the samples of 30 trees each, were applied to determine all the weights of the trees in the two cut areas and then to estimate the relative total fresh biomass present. The polynomial equation was chosen because it was the one that best represented the data set with the highest statistical coefficient of determination. To determine the moisture content of the trees and consequently the dry biomass produced at the time of cutting, a total of 20 trees were sampled, 10 for each section. Three 3 cm thick wooden discs were taken, in basal, medium, and high positions, for a total of 60 cylinders. The moisture content was determined by using a digital scale (accuracy 0.01 g), placing the samples to dry in a forced ventilation stove at a temperature of 105 ± 2 °C, according to the requirements of the European Standard EN ISO 18134-1:2015. The average value of moisture content obtained, distinguished for the two sections, was applied to determine the corresponding dry matter (DM) produced per year by two plantations.

### 2.2. Felling Machine and Working System

The machine used to fell the two areas was a Volvo EC 140 CL crawler excavator, with an engine power of 69 kW (Figure 1a), equipped with a Westtech C350 accumulating shear head (Figure 1b) with a standard cutting capacity of 35 cm. The total mass of the machine was 17,900 kg (including 1200 kg of the shear head), with an overall size of 2.53 m wide, 8.26 m of long, and 2.98 m of high. On the upper part of the shear head, there was one

collecting arm, which closed on a fixed arm, clutching the trunk; in addition, two other mobile pincer arms (accumulator arms) were set above to hold the already cut trees before proceeding to other cuts.

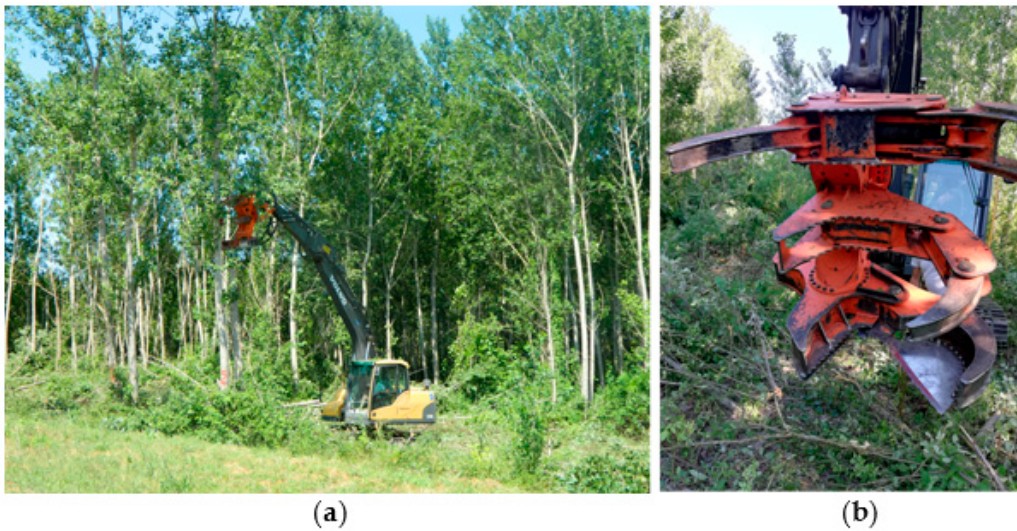

**Figure 1.** Crawler excavator during the felling of a 11-year-old tree (**a**) and details of the shear head (**b**).

The harvesting method applied was WTS. In the T8 plantation, the operator made a single cut at the base of the tree, moving and stacking it on the ground. In the T11 plantation, on the bigger trees, for a total of 152 trees (33%), he made two cuts: the first one at a height of about 6 m, moving and stacking this first part of the tree on the ground and the second one at the base of the tree. The machine, in both the T8 and T11 plantations, proceeded to cut 3 or 4 rows at a time.

*2.3. Time Study*

The time study was aimed at determining the productive working time (*PT*) of the felling operation including only delays (*D*) not exceeding 15 min for each event. Delays represent work interruptions and have been divided into mechanical (*MD*), i.e., service or repair, and personal (*PD*), such as interruptions due to the operator, including rest breaks [32].

The productive work carried out by the machine was divided into four main work phases, including a phase of movement and approach to the tree; a subsequent phase in which the shear head grabs and cuts the tree; a phase in which raises, moves, aligns, and stacks the tree; and a fourth phase concerning the cleaning of the ground from shrubs and dead trees to facilitate the approach to the next tree to be cut. In particular, the basic time elements considered were therefore the following:

- Moving time (*MT*), spent to move and position the machine in the place before felling;
- Felling time (*FT*), required for tree grabbing and cutting;
- Bunching time (*BT*), required for lifting, moving, and stacking the tree;
- Clearing time (*CT*), used for removing brambles, shrubs, and dead trees around the trees to be felled.

In the T11 section, on trees subjected to double cut as mentioned above, *FT* and *BT* represent the sum of the respective two times measured. Since *CT* and *D* are not related to tree characteristics and occur sporadically, the total time observed for these two elements was homogeneously attributed to each tree (total time divided per number of trees felled).

The time study follows the indications of the Forest Work Study Nomenclature suggested by IUFRO Working Party S3.04.02 [33] and approved by the IUFRO Division 3. The working times were detected using a Minerva chronometric table with three centesimal stopwatches, applying the snap-back timing method [27] and making videos for further checking of the working times observed during some work phases. The machine was observed at work for a total of 16.5 h, while the length of the videos was about 250 min.

The time study, distinct in the single work phases, was conducted with reference to each cutting cycle. Data were collected for the construction of a statistical model representative of the productive time (*PT*) as a function of the weight of the trees.

The model chosen that best interpolates the data referring to both sections T8 and T11 is described by the power equation (Equation (2)), whose general formula is the following:

$$PT = a \times W^b \tag{2}$$

where *PT* is expressed in $s\ tree^{-1}$ and *W* in $kgw\ tree^{-1}$ and *a* and *b* represent the coefficients of the equation.

*2.4. Productivity and Cost Evaluation Models*

On the basis of *PT* values recorded for each tree in the two sections, the work productivity per hour (*PH*) was determined [27,34] by applying Equation (3):

$$PH = 6 \times W/PT \tag{3}$$

where *PH* is expressed in $Mg\ h^{-1}$, W is the tree weight (kgw), and *PT* is the productive work time (centesimal seconds).

The related costs were also calculated on the basis of the *PH* thus obtained. The unit cost was calculated using the following Equation (4):

$$CMg = HC/PH \tag{4}$$

where *CMg* represents the unit cost ($EUR\ Mg^{-1}$) and *HC* is the machine hourly cost ($EUR\ h^{-1}$).

The following Equation (5) was instead used to calculate the cost per hectare:

$$CHa = CMg \times W/1000 \times N \tag{5}$$

where *CHa* is the cost per hectare ($EUR\ ha^{-1}$), *CMg* is the unit cost ($EUR\ Mg^{-1}$) determined by Equation (4), *W* is the tree weight (kgw), and *N* is the number of trees per hectare ($trees\ ha^{-1}$).

Once the data set was built, the productivity and cost evaluation models were determined for the T8 and T11 sections, adopting the equations derived as a function of the tree weight.

To determine the economic models, it was necessary to calculate the hourly cost of the felling machine, applying the analytical methodology proposed by Miyata [35]. The main technical-economic elements used in the calculation and the relative hourly costs are shown in Table 1. The hourly cost of the machine was used in Equation (4) to evaluate the unit cost for each tree cut and, subsequently, to elaborate the univariate regression models to estimate the cost per Mg and per hectare as a function of the tree weight.

**Table 1.** Main economic elements considered to calculate the hourly cost of the machine.

|  | Value |
|---|---|
| Purchace price (EUR) | 130,000.00 |
| Salvage value (EUR) | 10,833.00 |
| Life time (years) | 12 |
| Average Annual Investment (EUR year$^{-1}$) | 75,382.00 |
| Total productive time (h) | 14,400 |
| Scheduled hours (h year$^{-1}$) | 1600 |
| Productive hours (h year$^{-1}$) | 1200 |
| Fuel consumption (L h$^{-1}$) | 13.03 |
| Lubricants consumption (L h$^{-1}$) | 0.34 |
| Fuel price (EUR L$^{-1}$) (*) | 0.90 |
| Lubricant price (EUR L$^{-1}$) | 9.00 |
| Load engine factor (%) | 65 |

(*) Subsidized fuel price for the agricultural sector.

*2.5. Statistical Analysis*

Statistical data processing was performed using SPSS statistical software. The analysis was aimed at highlighting statistically significant differences between the T8 and T11 plantations in relation to the parameters examined and to build models about the performance and costs of the felling operation.

On the basis of the experimental data collected in the two plantations (DBH and heights), models were developed for the estimation of the regression curves of the weights of the trees as a function of the DBH. The weights calculated in this way were used for the construction of predictive models for estimating time, productivity, and costs (unitary and per hectare) distinct for the two plantations to compare them. To verify the condition of normality of the data and any significant difference between the two plantations in relation to the different variables considered, the Shapiro–Wilk test and the non-parametric Mann–Whitney test were performed, respectively.

**3. Results**

*3.1. SRC Poplar Plantations*

The statistical analysis of the dendrometric data showed non-normal distributions of the diameters and heights in the T8 and T11 sections (Shapiro–Wilk normality test, $p < 0.05$). Graphical representations of the characteristics of the distributions are shown in the box plots of Figure 2. The Mann–Whitney non-parametric test for two independent samples reveals the existence of statistically significant differences (at a level $p < 0.05$) in DBH, heights and weights, between the two groups. The results of the processing related to dendrometric measurements and those regarding the moisture content of the trees sample are indicated in Table 2. The harvested biomass was 31.15 Mg in T8 and 46.65 Mg in T11, corresponding to 6.65 and 9.25 Mg ha$^{-1}$ year$^{-1}$ of DM, respectively.

The relationship between the observed tree heights and diameters in the two sections is shown in Figure 3a, while the relationship between the weight and the diameter of the trees sample used to determine the equations of the weights as a function of DBH is reported in Figure 3b. The analysis of variance (ANOVA) of the four regression models reported in the figure produced a statistically significant result ($p < 0.01$), showing that the model adopted is good.

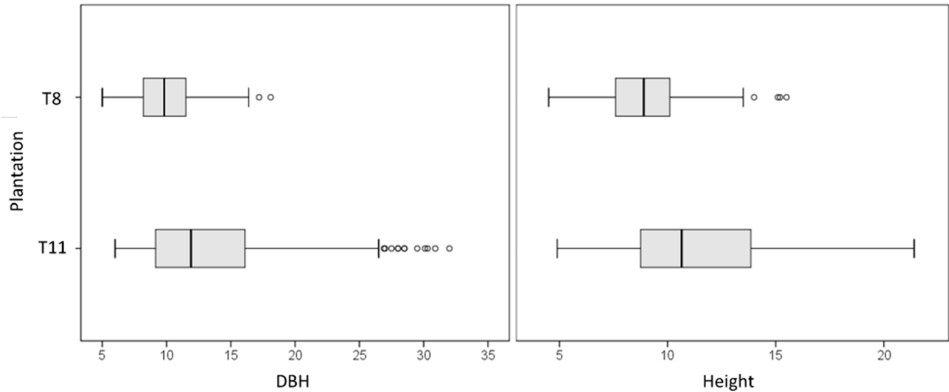

**Figure 2.** Box plots showing the comparison of the distributions of diameter at breast height (DBH) (cm) and height (m) in plantations T8 and T11.

**Table 2.** Dendrometric and productive characteristics of the T8 and T11 plantations (±SD in brackets).

|  | T8 | T11 |
| --- | --- | --- |
| Surface (ha) | 0.27 | 0.22 |
| Trees felled (N.) | 616 | 460 |
| Trees ha$^{-1}$ (N.) | 2281 | 2091 |
| Stumps failure (%) | 68.05 | 71.4 |
| DBH average (cm) | 9.84 (±2.30) | 13.20 (±5.51) |
| Height average (m) | 8.82 (±1.78) | 11.39 (±3.79) |
| Weight average (kgw) | 50.57 (±18.82) | 101.41 (±87.48) |
| Biomass (Mg) | 31.15 | 46.65 |
| Biomass (Mg ha$^{-1}$) | 115.37 | 212.05 |
| Biomass (Mg ha$^{-1}$ year$^{-1}$) | 14.42 | 19.28 |
| Moisture content (%) | 53.87 (±2.29) | 52.02 (±2.04) |
| Dry biomass (Mg$_{DM}$ ha$^{-1}$ year$^{-1}$) | 6.65 | 9.25 |

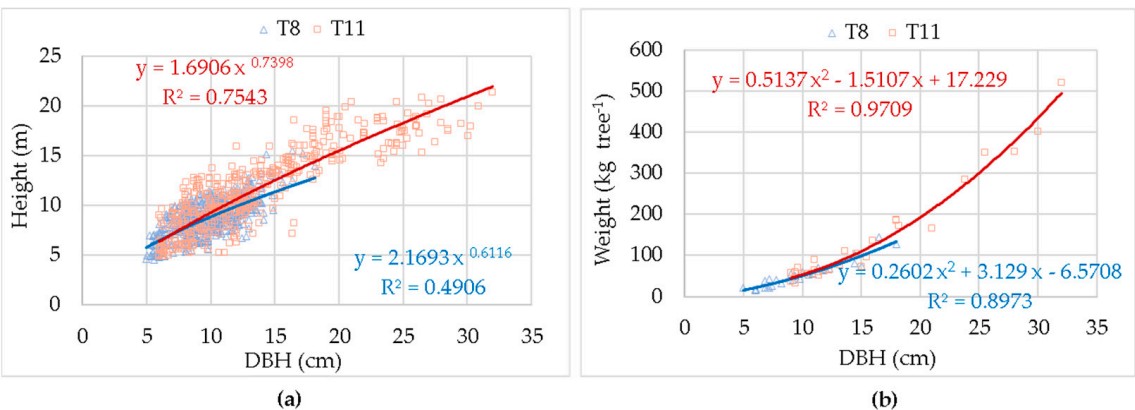

(**a**)  (**b**)

**Figure 3.** Relationship between height and diameter at breast height (DBH) in T8 (616 obs.) and T11 (460 obs.) plantations (**a**), and weight equations as a function of the DBH (30 obs. for each section) (**b**).

### 3.2. Time Study

Figure 4 shows the percentage value of working time per tree, broken down into individual elements. Figure 5 instead shows the comparison box plots of the distributions of the main elements of the working times in sections T8 and T11. The delay times were longer in T8, where the machine recorded frequent work stoppages due to technical problems regarding the shear hydraulic system. The incidence of *MD* was 9.2% in T8,

compared to just 2.2% in T11. The percentage of *PD* was similar in the two cases (4.6% and 4.2%), for a total delay of 13.8% and 6.4%, respectively.

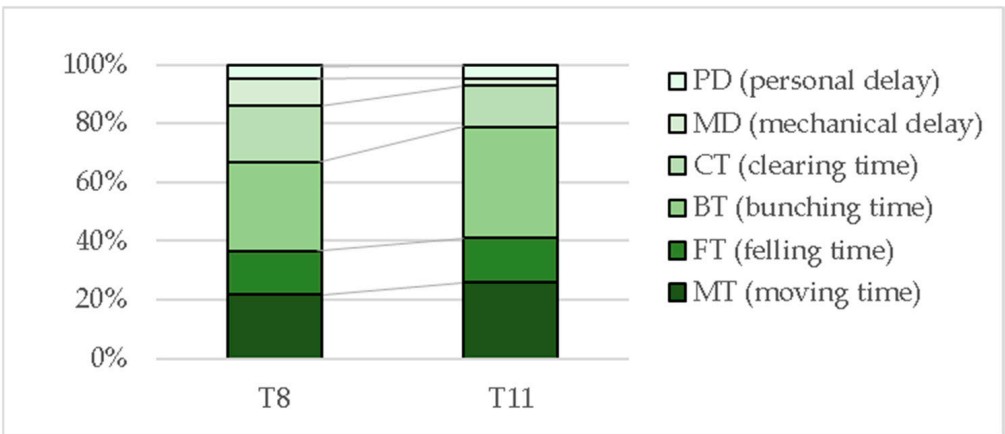

**Figure 4.** Percentage incidence of the single working phases per tree in plantations T8 and T11.

**Figure 5.** Box plots showing the comparison of *PT* (productive work time), *MT* (moving time), *FT* (felling time), and *BT* (bunching time) distributions in plantations T8 and T11 (in s tree$^{-1}$).

In general, a higher incidence of the main work phases was observed in T11, except for *CT* and *MD*. This is mainly because the machine made two cuts on about one-third of the total trees (bigger trees), increasing the average times per tree to perform the moving, felling, and stacking phases. The machine was observed for a total of 16.50 h, of which 52.2% was on T8 and was 47.8% on T11. Referring to the surface unit, the *PT* that was estimated as necessary was 31.83 h ha$^{-1}$ on the T8 section and 35.86 h ha$^{-1}$ on the T11 section.

The practice of two cuts per tree on the T11 section led to an increase in the average time consumed for the felling phase, which was 15.88 s tree$^{-1}$ ($\pm 6.67$ SD), compared to 12.46 s tree$^{-1}$ ($\pm 4.56$ SD) recorded in the T8 section. The total *PT* was 102.90 s ($\pm 21.61$ SD) and 83.83 s ($\pm 14.92$ SD), respectively. Statistical analysis using the Mann–Whitney test highlights statistically significant differences ($p < 0.05$) between the medians of each variable considered in sections T8 and T11. Figure 6 shows the two *PT* curves as a function of the tree weight. ANOVA test on the two curve models was statistically significant ($p < 0.01$). The anomalous trend of the *PT* curve on the plantation T8 and the low value of the determination coefficient ($R^2 = 0.106$) is justified by the fact that the operator showed greater difficulty in maneuvering on very small trees, with a consequent increase in *MT* and *BT*, compared to those found on bigger trees.

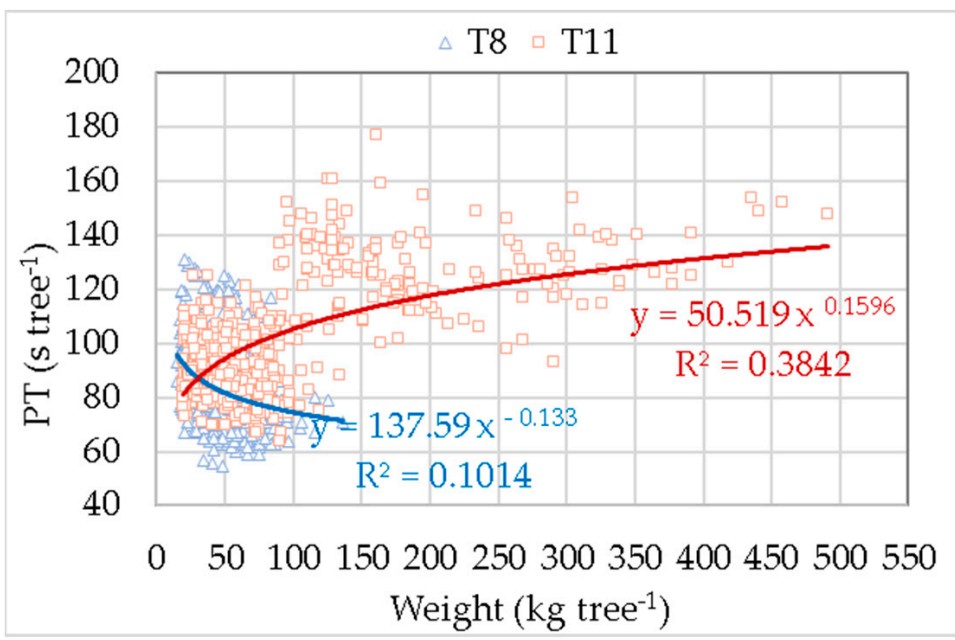

**Figure 6.** Productive work time (*PT*) as a function of the tree weight for plantations T8 and T11.

### 3.3. Productivity and Costs Evaluation Models

On the basis of *PT* data, productivity and costs were calculated for each tree. The hourly cost of the machine, calculated analytically, was 56.16 EUR h$^{-1}$, divided into 12.67 EUR h$^{-1}$ of fixed cost, 22.49 EUR h$^{-1}$ of variable cost, and 21.00 EUR h$^{-1}$ of driver cost. The distributions of productivity and costs data are compared in Figure 7. Statistically significant differences ($p < 0.05$) were found between groups T8 and T11.

Figures 8–10 report the curves of the productivity and cost evaluation models, for the two SRC plantations. In particular, the estimate models regard *PH* (Figure 8), *CMg* (Figure 9), and *CHa* (Figure 10). For all regression models shown in the figures, the ANOVA test performed statistically significant results with a *p*-value < 0.01.

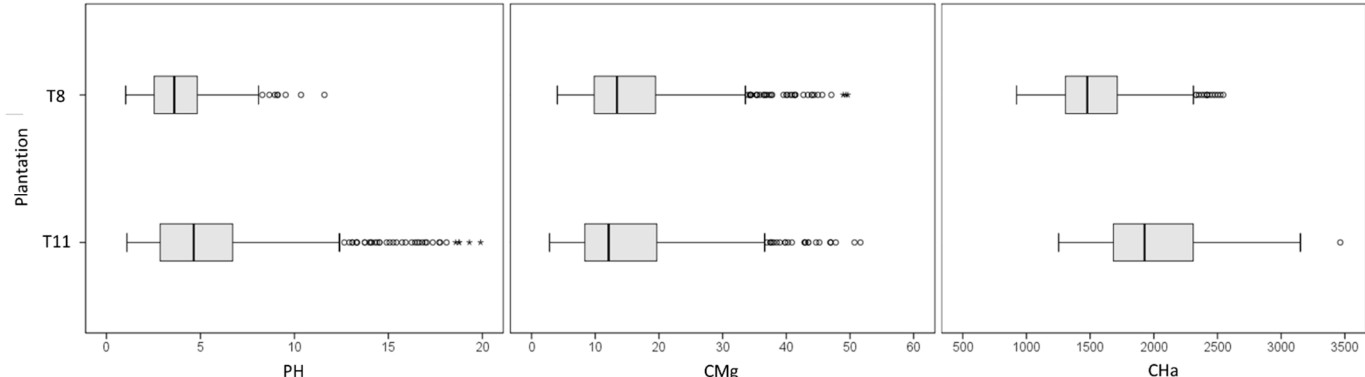

**Figure 7.** Box plots showing the comparison of data distributions of *PH* (work productivity), *CMg* (unit cost), and *CHa* (cost per hectare) in plantations T8 and T11.

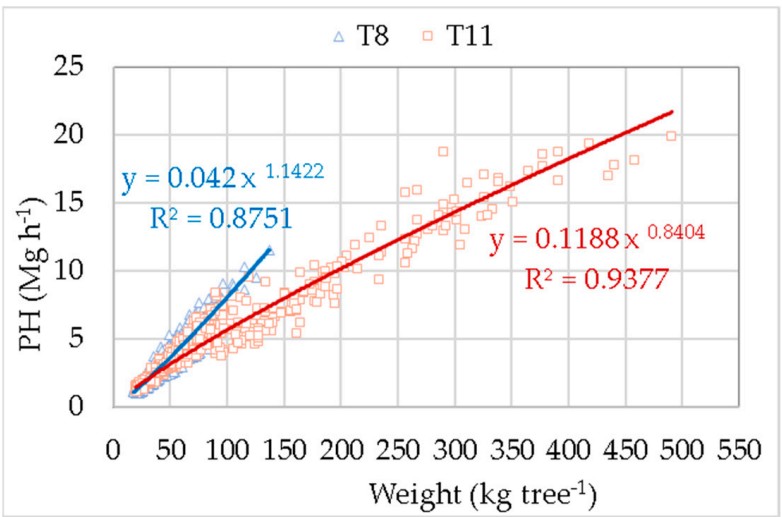

**Figure 8.** Work productivity (*PH*) estimate models calculated in relation to the weight of the tree for plantations T8 and T11.

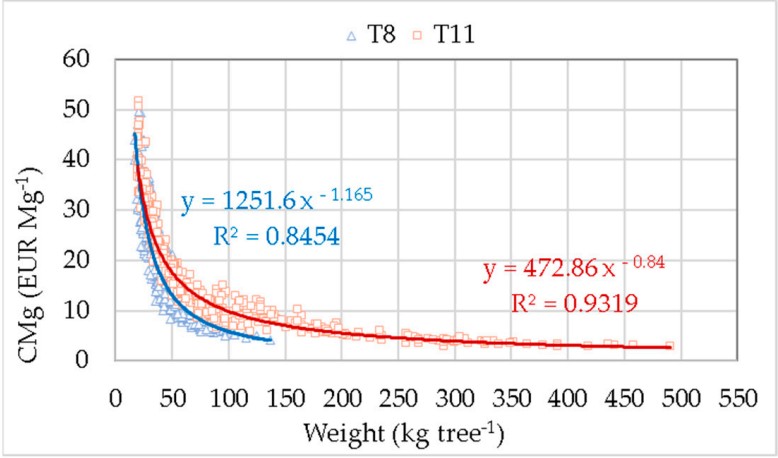

**Figure 9.** Estimation models of the unit cost (*CMg*) in relation to the weight of the tree for plantations T8 and T11.

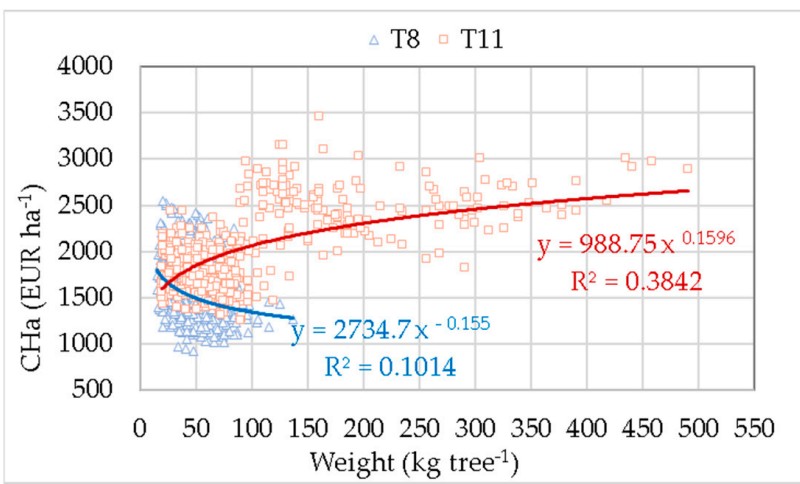

**Figure 10.** Cost per hectare (*CHa*) estimation models in relation to the weight of the tree for plantations T8 and T11.

Generally, the *PH*, as expected, increases with tree weight gain. The anomalous trend of the curves in T8 is justified by the greater working time spent on small trees, due to the greater difficulty of the machine in grabbing the trees and stacking them on the ground once cut. Referring to *PH* of section T8, the estimation model returns values between 1.02 to 11.60 Mg h$^{-1}$, when the weight of the tree varies from 17 to 137 kgw, with an average value of 3.80 ($\pm$1.71 SD) Mg h$^{-1}$. In section T11, it increases from 1.09 to 19.90 Mg h$^{-1}$, when the weight of the tree varies from 20 to 491 kgw, with an average value of 5.56 ($\pm$3.88 SD) Mg h$^{-1}$. The unit cost varies from 4.05 to 49.65 EUR Mg$^{-1}$ (average 15.98 $\pm$ 8.68 SD) and from 2.82 to 51.63 EUR Mg$^{-1}$ (average 15.31 $\pm$ 10.05 SD), corresponding to EUR 922.49–2545.11 ha$^{-1}$ (average 1542.10 $\pm$ 318.60 SD) and EUR 1252.17–3463.78 ha$^{-1}$ (average 2013.89 $\pm$ 423.02 SD), in T8 and T11 respectively.

## 4. Discussion

The development of SRC poplar plantations for energy purposes in Italy has suffered a setback in recent years, and part of the current surface has been converted by farmers to other crops considered more profitable (in central and southern Italy they have been replaced mainly with hazelnut). The remaining SRC plantations, in many cases, are not harvested, pending more favorable market conditions for biomass. This involves an increase in the size of the trees, and the harvesting cannot be carried out with traditional dedicated forage-based harvesters: it is necessary to use a typical forestry mechanization. Harvesting technology is evolving towards more advanced mechanization [36], which leads to an increase in work productivity and a reduction in costs [37]. Furthermore, and this is a fundamental aspect, the highest mechanization level in harvesting operations contributes to reducing both the severity and frequency of accidents and occupational diseases [38].

Referring to the SRC poplar plantations, it was found that the yields were not very high because these were managed from the beginning with a reduced energy input, with fertilization of the soil and water supply carried out only in the first years and reduced mechanical weeding interventions over the years. The limited yields are also attributable to the progressive increase over the years in the mortality of the stumps within the observed areas, which has reached a share of about 70%. This is also due to the climatic conditions typical of central Italy (dry summers), together with heavy and compact soil. The yields obtained were lower, especially in section T8, than previously observed on the same plantation harvested in 2- and 3-year cycles [10,31], as well as on other similar poplar plantations with longer cycles [14].

From an operational point of view, the felling machine encountered greater difficulties on the smaller trees of the T8 plantation compared to the T11, as shown in Figure 7. On

the other hand, even in the T11 plantation, the presence of larger trees led to an increase in working times as the operator was forced to make two cuts per tree, on a third of the plantation, also to facilitate the subsequent extraction of the trees by a forwarder. This has considerably increased the overall working times thanks to the doubling of the approaching, felling, and stacking phases. Furthermore, the operator, having no previous experience in harvesting this type of plantation, in both cases did not exploit the storage capacity of the shear, which would have allowed him to work more trees at the same time before depositing them on the ground, an operation that could have contributed to reducing working time. To this it must be added both the particularly high number of trees to be cut (over 2000 trees per hectare), and the small size of the tree compared to what can be found in a forest yard. These conditions did not allow reaching an average productivity like that observed in other forestry yards. In fact, the average work productivity obtained, about 4.7 Mg h$^{-1}$, was much lower than that recorded with similar machines and plantations by other authors [27,39–41], and more than halved compared to dedicated machines equipped with different felling heads [26,42].

The low average productivity achieved was also strongly influenced by the driver's lack of work experience on cutting these plantations. The subjective aspect in the work is very important and has also been highlighted in other studies concerning different types of mechanized harvesting on different plantations. It is difficult to quantify the driver's influence on final work productivity due to the simple fact that his training is not something that can be standardized globally [43,44]. Some of the studies carried out have shown that the technical preparation of the operator can affect up to 40% the final productivity in the use of advanced forestry equipment such as a harvester [44–48].

As clearly evidenced by the proposed cost model, the unit cost tends to be very high as the size of the trees becomes smaller. In this case, the machine does not work optimally and has a very low productivity, making the unit costs become unsustainable. On the other hand, as the trees size increases, the cost of the cutting operation decreases, reaching values that are acceptable and comparable with those observed by other authors [41].

In our study, as expected, hourly productivity increased as the weight of the trees increased. The range of values obtained varied from 1.02 to 11.60 Mg h$^{-1}$ for the T8 section and from 1.09 to 19.90 Mg h$^{-1}$, for the T11 section. The average productivity obtained was slightly higher in the T11 section with 5.6 Mg h$^{-1}$, compared to 3.8 Mg h$^{-1}$ in the T8 section. As regards unit costs, the differences between the two sections were more attenuated with a range of values between 4.05 to 49.65 EUR Mg$^{-1}$ in T8 and between 2.82 to 51.63 EUR Mg$^{-1}$ in T11. The average unit cost is similar in both cases, settling around 15 EUR Mg$^{-1}$ (15.98 EUR Mg$^{-1}$ for T8 and 15.31 EUR Mg$^{-1}$ for T11).

The cost-per-hectare model confirms the anomalous trend referred to as the T8 plantation. In this case, unlike the T11 curve, the cost increases as the tree size decreases due to the greater difficulty of the machine to work on smaller trees. However, the cost per hectare is always lower for the T8 plantation than for the T11, except when the weight of the trees becomes less than 25 kgw, where the break-even point occurs between the two plantations.

The economic sustainability of the use of shears on aged SRC poplar plantations must be evaluated considering that the felling operation represents only a part of the entire harvesting work, as the costs of extraction, chipping, and transport of the biomass must also be considered. This overall cost must then be compared with the market price of the woodchips, which is in any case currently low [14,49]. For this reason, the adoption of the mechanization examined for felling this type of plantation must be considered economically sustainable if its costs are contained within acceptable limits of about 15 EUR Mg$^{-1}$ for fresh biomass (about 30 EUR Mg$^{-1}$ DM).

According to the model we have adopted, therefore, the condition of economic sustainability, in the two 8- and 11-year-old SRC poplar plantations, occurs for trees weighing more than 45 and 60 kgw respectively.

Finally, the study highlighted the technical and economic difficulties of using advanced mechanization for the felling of aged SRC poplar plantations. However, the proposed models could still represent a useful tool for farmers to guide their decisions should they need to harvest SRC plantations under similar conditions.

**Author Contributions:** Conceptualization, G.S.; methodology, G.S.; formal analysis, G.S.; investigation, G.S., A.A., V.C., and A.D.G.; data curation, G.S.; writing—original draft preparation, G.S.; writing—review and editing, G.S., A.A., V.C., and A.D.G.; visualization, G.S.; funding acquisition, G.S. All authors have read and agreed to the published version of the manuscript.

**Funding:** This study was developed within the AGROENER "Energia dall'agricoltura: innovazioni sostenibili per la bioeconomia" national project, funded by Italian Ministry of Agriculture, Food and Forestry Policies (MiPAAF, D.D. n. 26329, 1 April 2016).

**Institutional Review Board Statement:** Not applicable.

**Informed Consent Statement:** Not applicable.

**Data Availability Statement:** Data is contained within the article.

**Conflicts of Interest:** The authors declare no conflict of interest.

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
