# Peer review of "Models for the Evaluation of Productivity and Costs of Mechanized Felling on Poplar Short Rotation Coppice in Italy"

_forests, doi:10.3390/f12070954_

Round 1
Reviewer 1 Report
The work presented gives interesting information and results to the forest and energy industries, forest owners and entrepreneurs especially in Italy. Unfortunately, the manuscript is not presented in a logical and concise manner. Material and methods are not presented clearly enough to allow other researchers to repeat the study. Conclusions are not justified and supported precisely by the information presented in the manuscript
Author Response
Point 1: The work presented gives interesting information and results to the forest and energy industries, forest owners and entrepreneurs especially in Italy. Unfortunately, the manuscript is not presented in a logical and concise manner. Material and methods are not presented clearly enough to allow other researchers to repeat the study. Conclusions are not justified and supported precisely by the information presented in the manuscript
Response 1:
We are very sorry that the work has aroused such critical views. We understand that the manuscript needs to be revised but there does not seem to be a lack of a logical and concise sequence in the topics covered.
We also have to disagree that "Materials and Methods" are not clear. It seems to us that the presentation of the data and the subsequent processing have been clearly expressed and treated in a logical and consequential way, from the characterization of the plantations to the description of the machine and the working system, to the study of the times, to finish with the predictive models proposed. We are sorry but general criticisms such as the one expressed, without any objective reference, do not help us to improve the content of the text.
The repetitiveness of the study depends on the conditions in which the work was carried out which, as clarified several times in the text (introduction, materials and methods, discussion), refers to the mechanized harvesting of SRC plantations on aged poplar. In these conditions, the dimensions reached by the trees no longer allow the use of dedicated machines (forage-based harvester). For this reason, the test involved the use of a machine equipped with a share head.
It was reiterated that the results can only be useful if they relate to conditions such as those described in the document.
Even the discussion and conclusions seem to us to correspond quite well to those expressed in the text.
However, we thank the reviewer for the observations which for us were however a precious element of reflection.
As suggested, the text has been revised as much as possible in the most critical points.
Reviewer 2 Report
The objective of the search as well as methodology are fairly acceptable. However there are a few points to be checked or improved. The followings are the detail:
1. Line 129, etc: The unit of "weight" should be "kgw" or "N" (Newton). If the authors have an intention of using "kg" as a unit, the word "mass" instead of "weight" should be used.
2. Lines 194-196, Equation (2): If the unit of PT is 0.01 sec., the equation should be "PH = 36 x W / (PT x 1000). It would be recommended that additional explanation on "centesimal seconds" or checking of the equation (2) is required.
3. Figure 5: In comparison with the contents of Figure 4, it seems that the component "CT" should be also included in the figure.
I hope that these comments will be some help for improving the manuscript.
Author Response
The objective of the search as well as methodology are fairly acceptable. However there are a few points to be checked or improved. The followings are the detail:
Point 1:
Line 129, etc: The unit of "weight" should be "kgw" or "N" (Newton). If the authors have an intention of using "kg" as a unit, the word "mass" instead of "weight" should be used.
Response 1:
As suggested, the text have been changed by replacing "kg" with "kgw".
Point 2:
Lines 194-196, Equation (2): If the unit of PT is 0.01 sec., the equation should be "PH = 36 x W / (PT x 1000). It would be recommended that additional explanation on "centesimal seconds" or checking of the equation (2) is required.
Response 2:
We are sorry but we cannot accept the suggestion, confirming the formula expressed in the text: PT represents the time for the felling of a tree, in centesimal seconds (s) (100 seconds=1 minute). For the transformation into hours, PT must at first be divided by 100, thus obtaining the minutes, and then further dividing by 60, to obtain the hours (1 hour=60 minutes). This means that, to get the hours, you need to divide PT by 6000.
The weight of the tree expressed in kgw, to be transformed into Mg, must be divided by 1000. Hence the formula: PH (Mg h-1) = (kgw/1000)/(s/6000) = (kgw/s) x (6000/1000) = 6 x (kgw/s) = 6 x W/PT.
Point 3:
Figure 5: In comparison with the contents of Figure 4, it seems that the component "CT" should be also included in the figure.
I hope that these comments will be some help for improving the manuscript.
Response 3:
Clearing Time (CT), even though representing a significant part of the total working time in absolute and percentage terms, this item was not included in Figure 5, as well as delays (D), because these two events occurred sporadically, producing few data that cannot be statistically processed. For this reason, for the purposes of the work and the determination of the models, the sum of the times for each was made, the average was then obtained, and this value, as explained in the text, was uniformly attributed to each observation per tree.
We thank the reviewer for suggestions which were very helpful in improving the manuscript.
Reviewer 3 Report
This work analyses the economical possibilities of mechanical harvesting of short rotation coppices (SRC) that must be let on the field for more years than projected, debt to the low prices and subsides of woodchips, that makes not interesting SRC harvesting at the age of 2-3 years, task than can be done with modified forage harvesters.
When the crops are harvested at the age of 8 years or more, forage harvesters are not suitable and it is necessary to use forest machinery.
Theoretical models of harvesting costs, based in experimental measures of RSC dendrometric variables, harvesting times and machinery costs, performed with forest specialized machinery, of crops 8 and 11 years old, was done.
The results show how there is a high variability of situations, and in function of that, the economical profitability of this task can or not be interesting. The study highlighted the technical and economic difficulties of using advanced mechanization for the felling of aged SRC poplar plantations. The proposed models could represent a useful tool available to guide the decisions of farmers who will have to proceed with the harvesting of SRC plantations under similar conditions.
As the models are based in experimental data, are of great interest to do further prospections.
The work is well structured, all the methods and results are clearly explained, and the discussion justify the main conclusions, consequently, this work is suitable to be published in the journal.
Author Response
Point 1: This work analyses the economical possibilities of mechanical harvesting of short rotation coppices (SRC) that must be let on the field for more years than projected, debt to the low prices and subsides of woodchips, that makes not interesting SRC harvesting at the age of 2-3 years, task than can be done with modified forage harvesters.
When the crops are harvested at the age of 8 years or more, forage harvesters are not suitable and it is necessary to use forest machinery.
Theoretical models of harvesting costs, based in experimental measures of RSC dendrometric variables, harvesting times and machinery costs, performed with forest specialized machinery, of crops 8 and 11 years old, was done.
The results show how there is a high variability of situations, and in function of that, the economical profitability of this task can or not be interesting. The study highlighted the technical and economic difficulties of using advanced mechanization for the felling of aged SRC poplar plantations. The proposed models could represent a useful tool available to guide the decisions of farmers who will have to proceed with the harvesting of SRC plantations under similar conditions.
As the models are based in experimental data, are of great interest to do further prospections.
The work is well structured, all the methods and results are clearly explained, and the discussion justify the main conclusions, consequently, this work is suitable to be published in the journal.
Response 1:
We sincerely thank the reviewer for the substantially positive opinion on the manuscript. We welcome the suggestion of further investigation into this matter.
Round 2
Reviewer 1 Report
Is there actual need to make separate models for T8 and T11 plantations? I ask that you combine the data and draw conclusions about the economic sustainability on that basis.
Furthermore I hope that the time study as well as procedure of productivity and cost evaluation models are reported more detailed.
Author Response
Point 1:
It there actual need to make separate models for T8 and T11 plantations? I ask that you combine the data and draw conclusions about the economic sustainability on that basis.
Response 1:
Treating the data as a single sample would not be a problem, the calculation simulations would lead, as evident, to almost intermediate results compared to the data treated separately. The equations in this case chosen as the best model would largely consist of quadratic equations rather than power equations. The purpose of the work, on the other hand, was precisely to detect and compare the differences between two different conditions of the SRC poplar plantation, in terms of felling machine performance, economic costs and relative economic sustainability of the operation. The reviewer's suggestion would completely overturn the meaning of the study that relies on comparison, so unfortunately, we are not inclined to accept this suggestion.
Point 2.
Furthermore I hope that the time study as well as procedure of productivity and cost evaluation models are reported more detailed.
Response 2:
As suggested, more details have been included in the text with reference to the time study (line 190).